# Deep Learning Algorithms for Estimation of Demographic and Anthropometric Features from Electrocardiograms

**DOI:** 10.3390/jcm12082828

**Published:** 2023-04-12

**Authors:** Ji Seung Ryu, Solam Lee, Yuseong Chu, Sang Baek Koh, Young Jun Park, Ju Yeong Lee, Sejung Yang

**Affiliations:** 1Department of Precision Medicine, Yonsei University Wonju College of Medicine, Wonju 26426, Republic of Korea; ryujissss@yonsei.ac.kr; 2Department of Preventive Medicine, Yonsei University Wonju College of Medicine, Wonju 26426, Republic of Korea; esolami@naver.com (S.L.); kohhj@yonsei.ac.kr (S.B.K.); 3Department of Dermatology, Yonsei University Wonju College of Medicine, Wonju 26426, Republic of Korea; ljy0302@naver.com; 4Department of Biomedical Engineering, Yonsei University, Wonju 26493, Republic of Korea; dbtjd9967@naver.com; 5Division of Cardiology, Department of Internal Medicine, Wonju Severance Christian Hospital, Yonsei University Wonju College of Medicine, Wonju 26426, Republic of Korea; pyj@yonsei.ac.kr

**Keywords:** electrocardiography, demographics, deep learning, artificial intelligence, age, sex, BMI, blood type

## Abstract

The electrocardiogram (ECG) has been known to be affected by demographic and anthropometric factors. This study aimed to develop deep learning models to predict the subject’s age, sex, ABO blood type, and body mass index (BMI) based on ECGs. This retrospective study included individuals aged 18 years or older who visited a tertiary referral center with ECGs acquired from October 2010 to February 2020. Using convolutional neural networks (CNNs) with three convolutional layers, five kernel sizes, and two pooling sizes, we developed both classification and regression models. We verified a classification model to be applicable for age (<40 years vs. ≥40 years), sex (male vs. female), BMI (<25 kg/m^2^ vs. ≥25 kg/m^2^), and ABO blood type. A regression model was also developed and validated for age and BMI estimation. A total of 124,415 ECGs (1 ECG per subject) were included. The dataset was constructed by dividing the entire set of ECGs at a ratio of 4:3:3. In the classification task, the area under the receiver operating characteristic (AUROC), which represents a quantitative indicator of the judgment threshold, was used as the primary outcome. The mean absolute error (MAE), which represents the difference between the observed and estimated values, was used in the regression task. For age estimation, the CNN achieved an AUROC of 0.923 with an accuracy of 82.97%, and a MAE of 8.410. For sex estimation, the AUROC was 0.947 with an accuracy of 86.82%. For BMI estimation, the AUROC was 0.765 with an accuracy of 69.89%, and a MAE of 2.332. For ABO blood type estimation, the CNN showed an inferior performance, with a top-1 accuracy of 31.98%. For the ABO blood type estimation, the CNN showed an inferior performance, with a top-1 accuracy of 31.98% (95% CI, 31.98–31.98%). Our model could be adapted to estimate individuals’ demographic and anthropometric features from their ECGs; this would enable the development of physiologic biomarkers that can better reflect their health status than chronological age.

## 1. Introduction

The electrocardiogram (ECG) is a noninvasive piece of diagnostic equipment that records physiological activities over time. It has become the standard tool for diagnosing arrhythmia and ischemic heart disease. Furthermore, an ECG analysis can also aid in the detection of electrolyte abnormalities such as hyperkalemia [1]. ECGs can be collected from homes [2,3], portable devices [4,5], and hospitals, expanding their applicability in various situations.

With recent developments in technology and computing power, deep learning algorithms are being used in medicine for disease diagnosis and prognostic stratification [6]. Deep learning algorithms are a type of artificial intelligence that can recognize complex patterns and features in large datasets during the learning process. Due to this capability, these algorithms can be applied to various fields to facilitate data analysis and decision-making processes. They are primarily applied in imaging modalities, such as magnetic resonance imaging, computed tomography, plain radiography, ultrasonography [7], pathology imaging [8], and clinical imaging of dermatologic disorders. In contrast, deep learning can be applied not only to high-dimensional images but also to low-dimensional data. It is also utilized in the analysis of other biosignals that measure microcurrents within the body, such as ECGs [9,10], electroencephalograms (EEGs), and electromyograms (EMGs). These signals provide important information about the activity of the brain and muscles, respectively. The application of ECGs to deep learning algorithms has been extensively attempted in the past few years [11,12]. Most studies have focused on processing ECG signals, such as feature extraction or noise reduction, or mainly on classifying abnormalities or types of single heartbeats [13,14,15]. Recently, beyond the existing analysis, various approaches and attempts have been made to determine whether deep learning algorithms can predict diseases using ECGs [16,17,18].

Such an approach enables the utilization of various ECG features that have not previously been identified through the human eye or the traditional rule-based approach for interpreting ECGs, leading to more accurate predictions in a broader array of diseases [19]. We postulated that being able to predict subjects’ demographic and anthropometric features from their ECGs would enable the development of physiologic biomarkers that can better reflect their health statuses than chronological age [20], as well as reduce various types of human errors that can arise while entering subject information during ECG acquisition [21,22]. Some studies have reported significant differences in QRS duration depending on age [23]. QT interval characteristics, ST segment, and others can contribute to gender identification [24]. A pathological basis can be seen to characterize an individual through ECG changes. Previous studies have utilized artificial intelligence to predict demographic characteristics such as age and sex based on ECG data [25]. While there have been studies that have identified an association between BMI and ECG components, none have employed regression and classification techniques [26,27]. To our knowledge, there are currently no studies on predicting the ABO blood type using ECG data. In light of these gaps in the literature, we designed an artificial intelligence prediction model to address these issues and provide a novel approach to ECG-based prediction. Our study aims to provide new insights and references for predicting demographic and anthropometric features using ECG data.

This study aimed to develop and validate a deep learning model that uses ECGs to predict various demographic and anthropometric features, such as age, sex, ABO blood type, and body mass index (BMI). The objective of this study was to develop and validate a deep learning model that utilizes electrocardiograms (ECGs) for predicting several demographic and anthropometric features, including age, sex, type, body mass index (BMI), and ABO type. To achieve this goal, the study employed ECG big data from Yonsei University’s Wonju Severance and deep learning algorithms. The findings of this study confirm our assumptions and demonstrate the potential for evaluating individuals in greater detail. This approach offers a comprehensive assessment of an individual’s overall health in the medical field. Additionally, precise predictions of demographic and anthropometric features can prevent long-term or potential heart disease by assessing ECG function and reducing errors. The findings of previous studies demonstrate that it is possible to evaluate individuals in greater detail, which can be a valuable approach in the medical field. By taking into account an individual’s overall health status, an accurate prediction of demographic and anthropometric features can be achieved through an ECG analysis. This can ultimately help prevent potential long-term heart diseases by improving the accuracy of ECG-based assessments.

## 2. Materials and Methods

### 2.1. Data Collection

This retrospective study included individuals aged 18 years or older with at least one standard supine 10 s 12-lead 500 Hz ECG acquired in Yonsei University Wonju Severance Christian Hospital, Wonju, Korea, from October 2010 to February 2020 (Figure 1). This included both patients with or without cardiac diseases and healthy individuals who visited for a health examination. Demographic information, including age, sex, ABO blood type, and BMI, was collected via a retrospective chart review. Digital ECGs were directly exported from the MUSE Cardiology Information System (GE Healthcare, Chicago, IL, USA). The original source contained eight leads (lead I, II, V1, V2, V3, V4, V5, and V6). The rest of the four leads (III, aVF, aVR, and aVL) were added through vector calculation. Leads I, II, and III comprised Einthoven’s triangle, which can be regarded as a closed circuit. Therefore, lead III was calculated using Kirchhoff’s law for two leads (I and II). The aVR, aVF, and aVL leads, known as the Goldberger leads, were composed of the same electrodes as the leads of Einthoven. Consequently, three leads (aVR, aVF, and aVL) can be calculated with three leads (I, II, and III) using Goldberger’s equation [28]. This vector calculation method is widely used [29]. The study was approved by the Institutional Review Board of Yonsei University Wonju Severance Christian Hospital and conformed to the ethical guidelines of the Declaration of Helsinki (approved on 18 January 2020, approval number CR319173). Because we anonymized the dataset by eliminating information that can be used to identify individuals, such as birthday (except birth year for the calculation of age variable), address, and telephone numbers, a waiver of written informed consent was granted for de-identified data.

### 2.2. Dataset

For the age, sex, and ABO blood type prediction models, subjects with at least one previous instance of ABO blood typing were included in the dataset (Figure 1). When multiple ECGs were available for a single subject, only the initial ECG was used to train and validate the models. Exceptional cases, such as changes in sex or blood type due to hematopoietic stem cell transplantation during the study period, were not considered for data curation. Unlike the other variables, the BMI was only available for some subjects, as the information was only collected when performing cardiac ultrasonography. Therefore, in most cases, the BMI and ECG measurements were within one month (31 days). When multiple ECGs were available for a single subject, the ECG closest to the BMI measurement was used to train and validate the BMI prediction model.

For all tasks, the dataset was partitioned into a training set (40%), validation set (30%), and test set (30%). Partitioning was performed so that the class balance was kept consistent for the respective task of each deep learning model. Partitioning was performed on an individual subject basis, and there was no overlap between the sets. The training and validation sets were used to determine and tune the parameters while training the model. Each model with the highest performance in the validation set was evaluated for its final performance using a test set.

### 2.3. Classification and Regression Task

The following four variables were to be predicted by the models (Figure 2 and Figure 3): age, sex, ABO blood type, and BMI. The prediction model used two independent models: classification and regression. Binary classification was conducted to verify that the classification model could classify subjects according to variations, and the regression model was used to perform the prediction. For the binary classification of age, 40—a numerical value that generally divides young and older subjects—was used as a cut-off, and for the binary classification of BMI, 25—a criterion for dividing obesity—was used as a cut-off. The age prediction model was comprised of two models: a classification model that performed a binary classification of the subjects into those aged below 40 years and those aged 40 years or above, and a regression model that directly predicted the subjects’ ages. The sex prediction model was a classification model that classified the subjects into male and female. The ABO blood type prediction model was a multiclass classification model that classified the subjects into A, B, AB, and O blood types. The BMI prediction model was also comprised of two models: a classification model that classified the subjects into those with BMIs below 25 kg/m^2^ and those with BMIs 25 kg/m^2^ or above according to the Korean diagnostic criteria for obesity [30], and a regression model that directly predicted the subjects’ BMI.

### 2.4. Deep Learning Method

The deep learning model was based on the convolutional neural network (CNN) model widely used in imaging, and the layers have been modified to enable the ECG to be learned. Because the ECG used as an input was a one-dimensional (1D) signal, the amount of information on the pattern was relatively small compared to the image. Therefore, the model was designed to be simpler. The convolutional and pooling layers of the model were used to extract features from the ECG data. In other words, these layers enabled the extraction of ECG features. The fully connected layer was responsible for generating the final feature by calculating the representative values for each dimension at the end of the process. The model was constructed using three primary blocks and a fully connected layer. Each of the primary blocks contained four layers, which included a 1D convolution layer (with a kernel size of 5), a batch normalization layer, the ReLU activation function, and a 1D max pooling layer. The 12-lead models took 12 (leads) × 10 (seconds) × 500 (Hz) ECG data as input and extracted their features through the main block. In contrast, the 1-lead models took lead II (excerpt from standard 12-lead supine ECGs) as a single input lead × 10 (seconds) × 500 (Hz). The extracted features were then processed through the pooling and flatten layers, and they became a feature map with a form of 192 × 1. The features then passed through a fully connected layer consisting of three layers, where each layer had 192, 64, and 32 nodes. There were two output nodes in the classification models and one output node in the regression models. The final output layer of the classification model was selected as the softmax layer, which outputs the actual probability corresponding to the class. Accordingly, the second node corresponding to the positive class between the two nodes was used as the final output. As the final output, the classification models provided the probabilities of the input belonging to each class, and the regression models provided the directly predicted values. The cross-entropy and mean absolute error (MAE) loss were respectively used as the loss function for each type of model.

### 2.5. Scaling a Model

The hyperparameters were determined through repeated experiments and a grid search. In particular, the batch size, learning rate, and kernel size of the max pooling layer were considered potential candidate parameters. The batch sizes were considered to be in the range from 64 to 512, and the learning rate was considered to be between 0.0001 and 0.1. Within the deep learning model, the kernel size of the convolution layer was fixed with the parameters determined through the grid search. Subsequently, the kernel size of the max pooling layer was considered to be between 2 and 5, and the case with the highest ARUOC was optimally defined for the validation set.

### 2.6. Statistical Metrics

The performance of the classification models was evaluated with the area under the receiver operating characteristic (AUROC). Based on the prediction results of the deep learning model and the comparison between the actual labels, each of the ECGs was classified into true positive (TP), false negative (FN), true negative (TN), false positive (FP), and was used in the metrics calculation. Accuracy, sensitivity, specificity, positive predictive value, and negative predictive value were measured using the optimal cut-off with the receiver operating characteristic (ROC) curve calculated based on the maximized Youden index (J). The metrics were calculated as follows:(1)$$Accuracy=\frac{TP+TN}{TP+FN+TN+FP}$$
(2)$$Sensitivity=\frac{TP}{TP+FN}\;\;\;\;\;\;\;\;\;\;\;\;\;\;\;\;\;\;\;$$
(3)$$Specificity=\frac{TN}{TN+FP}\;\;\;\;\;\;\;\;\;\;\;\;\;\;\;\;\;\;\;$$
(4)$$PPV=\frac{TP}{TP+FP}\;\;\;\;\;\;\;\;\;\;\;\;\;\;\;\;\;\;\;\;\;\;\;\;\;\;\;\;\;\;\;\;\;$$
(5)$$NPV=\frac{TN}{TN+FN}\;\;\;\;\;\;\;\;\;\;\;\;\;\;\;\;\;\;\;\;\;\;\;\;\;\;\;\;\;\;\;$$
(6)$$J=Max_{C}\left(Sensitivity+Specificity-1\right),$$
where *C* is the cut-off point in the ROC curve.

The MAE, Pearson correlation coefficient (Pearson R), and intraclass correlation coefficient (ICC) were measured for the regression models to determine the difference between the observed and estimated values. Years in the age task and kg/m^2^ in the BMI task were set as the basic unit of MAE. Potential bias was evaluated using a Bland–Altman plot, in which the *x*-axis was the mean of the observed and predicted values, and the *y*-axis was the difference between them. The metrics were calculated as follows:(7)$$MAE=\frac{1}{N}\overset{N}{\underset{i=0}{\sum }}\left|y_{i}-{\ddot{y}}_{i}\right|,\;\;\;\;\;\;\;\;\;\;\;\;\;\;\;\;\;\;\;$$
where *y_i_* is an observed value for the *i*th ECGs, and *ÿ_i_* is an estimated value.

Because our dataset consisted of standard 12-lead supine ECGs, we set the performance of the 12-lead ECG-based models as the main outcomes of our study. However, the performance of the 1-lead models was also reported to investigate the impact of the multi-lead information. All outcomes were measured for the validation dataset and test dataset. All statistics data were reported as point estimates and 95% confidence intervals (CIs). Data were analyzed and visualized using Python 3.8.5 (Python Software Foundation).

### 2.7. Visualization for Model Explanation

Gradient-weighted class activation mapping (Grad-CAM) [31] is a visual representative method for interpreting the decision of the trained CNN model. Each channel of the input 12-lead ECGs was converted into an image by plotting it on a two-dimensional plane, and all channels were then added to make it an image for the Grad-CAM. Because our model was a 1D CNN model, the size of the extracted heat map was a one-line array. Therefore, after resizing the image size through interpolation, the Grad-CAM was completed by combining the previously created image. The red-to-yellow area of the heat map created using the Grad-CAM method significantly impacted the CNN’s decision; hence, the closer the area was to blue, the less impact it had. In this study, we analyzed the explanatory model using the Grad-CAM method for three classification tasks.

### 2.8. Data and Code Availability

The data used in this study cannot be disclosed without the permission of the Ethics Committee (irb@yonsei.ac.kr, 82-033-741-1715). The data contain potentially sensitive information such as the subject’s date of birth and gender. Thus, the data are not publicly available because of privacy or ethical restrictions. All codes used for model development, analysis related to the current submission, and future updates will be available in the following repository: (https://github.com/RyuJiSSSS/ECG-Classification-for-YMJ-task.git) (accessed on 6 July 2022).

## 3. Results

### 3.1. Study Population

A total of 124,415 subjects were included throughout the study period (Figure 1). The age range of the subjects included in our study was 18 to 108 years, with a median age of 55 years. Among all subjects, 60,835 (48.90%) were female, and the mean age was 55.2 (SD, 17.3) years. The ECGs of 124,415 subjects were utilized in the training and validation of the age, sex, and ABO blood type prediction models (Figure 3 and Table 1). Among the ECGs, 49,762 (40%) were used as the training set, 37,324 (30%) as the validation set, and the remaining 37,329 (30%) as the test set. Meanwhile, the ECGs of 48,488 subjects were used in the training and validation of the BMI prediction model. These subjects had a mean BMI of 24.51 kg/m^2^ (SD, 14.17 kg/m^2^), with 20,402 (42.07%) having a high BMI (BMI over 25 kg/m^2^). Among the ECGs, 19,393 (40%), 14,546 (30%), and 14,549 (30%) were used as the training, validation, and test sets, respectively.

### 3.2. Evaluation Protocol

When experiments were conducted with ECGs with signal processing such as normalization or noise reduction, there was no significant difference between the results of each model. Therefore, this study conducted an experiment using raw ECGs for all tasks. The optimal batch size was 512, and the initial learning rate was 0.001. The max pooling kernel size of the deep learning model was set to 2. However, the difference in performance was minimal across the hyperparameter settings. The model was trained for 50 epochs using the Adam optimizer. During the learning process, the classification model was defined and stored as the best-performing model when the AUROC for the validation set was the highest. The regression model had the lowest MAE values. Subsequently, the stored best model was called, and final verification was performed using a test set. PyTorch version 1.10 was used as a deep learning framework.

### 3.3. Age Estimation

A performance summary of the classification model for age (<40 years vs. ≥40 years) is provided in Table 2 and Figure 4a. The model with the highest performance in the validation dataset achieved an AUROC curve of 0.923 (95% CI, 0.922–0.923) in the test dataset. The maximum Youden index had a sensitivity of 81.96% (95% CI, 81.55–82.37%) and a specificity of 86.67% (95% CI, 86.27–87.08%). The model with a 1-lead ECG showed a significantly inferior performance (AUROC, 0.816 [95% CI, 0.814–0.818]) to that of the model with the 12-lead ECG (AUROC, 0.923 [95% CI, 0.922–0.923]) (Table 3). A performance summary of the regression model for the direct prediction of age is provided in Table 2 and Figure 4b,c. The MAE between the predicted age and estimated age was 8.410 (95% CI, 8.380–8.441) years, with a Pearson correlation coefficient of 0.782 (95% CI, 0.780–0.784) and ICC of 0.636 (95% CI, 0.628–0.645). The Bland–Altman plot showed a proportional bias of −0.05. The model with a 1-lead ECG showed a significantly inferior performance (MAE, 8.410 [95% CI, 8.390–8.441]) to that of the model with a 12-lead ECG (MAE, 11.220 [95% CI, 11.096–11.344]).

### 3.4. Sex Estimation

A performance summary of the classification model for sex is provided in Table 2 and Figure 5. The model with the highest performance in the validation dataset achieved an AUROC curve of 0.947 (95% CI, 0.945–0.948) in the test dataset. The maximum Youden index had a sensitivity of 87.42% (95% CI, 85.92–88.91%) and a specificity of 86.25% (95% CI, 85.07–87.42%). The model with a 1-lead ECG showed a significantly inferior performance (AUROC, 0.807 [95% CI, 0.806–0.808]) to that of the model with a 12-lead ECG (AUROC, 0.947 [95% CI, 0.945–0.948]).

### 3.5. ABO Blood Type Estimation

A performance summary of the classification model for ABO blood type is provided in Table 2 and Figure 6a. The multiclass classification for the A, B, AB, and O blood types had a top-1 accuracy of 31.98% (95% CI, 31.98–31.98%). The binary classification model also had a mean AUROC curve of only 0.501 (95% CI, 0.496–0.506) (Figure 6b), with a mean sensitivity of 56.12% (95% CI, 4.96–107.27%) and a mean specificity of 44.03% (95% CI, −7.31–95.37%), showing a poor predictive power overall. The experiments were not repeated for the 1-lead models since even the 12-lead main model did not reveal any predictive power.

### 3.6. BMI Estimation

A performance summary of the classification model for BMI (<25 kg/m^2^ vs. ≥25 kg/m^2^) is provided in Table 2 and Figure 7a. The model with the highest performance in the validation dataset achieved an AUROC curve of 0.765 (95% CI, 0.763–0.766) in the test dataset. The maximum Youden index had a sensitivity of 70.00% (95% CI, 67.80–72.20%) and a specificity of 69.82% (95% CI, 67.30–72.35%). The model with a single-lead ECG showed a significantly inferior performance (AUROC, 0.765 [95% CI, 0.763–0.766]) to that of the model with a 12-lead ECG (AUROC, 0.633 [95% CI, 0.629–0.639]). A performance summary of the regression model for a direct prediction of BMI is provided in Table 2 and Figure 7b,c. The MAE between the predicted BMI and estimated BMI was 2.332 (95% CI, 2.328–2.336), with a Pearson correlation coefficient of 0.531 (95% CI, 0.530–0.533), and an ICC of 0.474 (95% CI, 0.464–0.484). The Bland–Altman plot showed a proportional bias value of 0.12. The model with a single-lead ECG showed a significantly inferior performance ECG (MAE, 2.684 [95% CI, 2.682–2.687]) to that of the model with a 12-lead ECG (MAE, 2.743 [95% CI, 2.726–2.760]).

### 3.7. Visualization for Explainable AI

Figure 8 shows the Grad-CAM for the age, sex, and BMI classification models. Grad-CAMs toward the old age class for ECGs acquired from young (Figure 8a) and old individuals (Figure 8b) are presented. Activation of the P wave, PR interval, and T wave was prominent only in the ECGs from old age. In Grad-CAMs toward the female class for ECGs acquired from male (Figure 8c) and female individuals (Figure 8d), activation of the T wave was only prominent in ECGs from female individuals. In Grad-CAMs toward a high BMI class for ECGs acquired from low BMI (Figure 8e) and high BMI individuals (Figure 8f), diffuse activation for diverse segments was only present in ECGs from high BMI individuals, where there was only activation on the QRS complex in ECGs from low BMI individuals.

## 4. Discussion

Our study confirmed that the analysis of ECGs using deep learning algorithms could predict anthropometric information, such as BMI, as well as demographic information, such as age and sex, with high accuracy. However, it showed that ECGs were not as useful in predicting ABO blood types. In addition, our study showed that the deep learning algorithm could reliably predict 1-lead ECGs, even if the accuracy was significantly lower than that of 12-lead ECGs.

Many previous studies have reported that age and sex affect ECGs [32,33,34]. For an individual, ECGs taken at a later age had more prolonged PR and QT intervals, shorter QRS durations, and diverse T wave amplitudes than ECGs taken at a younger age [35]. In addition, many studies have reported that ECGs differ by sex among healthy individuals [36]. Cases of a shorter PR interval and QRS duration, lower ECG voltage, longer QT interval, and more prevalent ST-segment changes following ischemic heart disease were more commonly observed in females than in males [37]. In our study, a Grad-CAM suggested that the model focused on P wave, PR interval, T wave, and diverse ECG segments in the prediction of demographic features from ECGs. Although we could not confirm that the model utilized the aforementioned features for prediction solely with the Grad-CAMs, the activation pattern suggests that the model localized the amplitude and position of the major ECG components.

A study on predicting age and sex using 12-lead ECGs was conducted prior to our study (Table 4) [25,38,39]. The deep learning model used in the study achieved an AUROC of 0.923 in age classification and an AUROC of 0.947 in sex classification, showing a much higher performance than classifications based on parameters traditionally extracted from ECGs [25]. As the study used a dataset and setting different from ours, the model’s performance could not be directly compared to ours. Notably, racial bias is also one of the primary considerations in analyzing ECGs with deep learning [40]; thus, differences in the racial compositions of the study population could also affect the results. The fact that most subjects in our study originated from an Asian population indicates that the predictions of age and sex using deep learning algorithms could be applied to subjects from various ethnic backgrounds.

BMI was another variable that we could effectively predict from the ECGs. It has previously been reported that obesity affects ECGs as well [26]. An increased P wave duration, an increased P wave, a leftward shift of the heart axis, and a low QRS voltage are commonly observed in a subject with a high BMI [27]. Furthermore, obesity is the most significant risk factor for left atrial enlargement [41], which is significantly related to an increased prevalence of atrial fibrillation and the occurrence of cardiovascular events and death. Although no clear mechanism has been found, metabolic and inflammatory functions arising from increased epicardial and pericardial fats may contribute to this. Although measuring BMI is simpler and easier in most circumstances than obtaining ECGs, in our study, ECGs were used to predict BMI, which may be helpful in patients for whom height and weight measurements are difficult. For instance, patients in the intensive care unit or those with unclear consciousness would benefit from this approach. The accuracy of our algorithm in predicting BMI was demonstrated by an AUROC of 0.765 and a MAE of 2.332 kg/m^2^, which were lower than those in predicting age and sex. However, as there have been no previous studies on BMI prediction using a machine learning analysis of ECGs, it was impossible to compare the performance of our algorithm with that of existing algorithms. In our study, we found that using ECGs to predict BMI can be particularly beneficial for patients for whom traditional height and weight measurements are difficult to obtain. This includes individuals in the intensive care unit or those with unclear consciousness, where alternative methods for assessing BMI may be necessary. Our findings highlight the potential of the ECG-based BMI prediction as a practical and effective approach in such scenarios.

ABO blood type was another variable that we aimed to predict in our study. However, we observed that ABO blood type was nearly unpredictable based on a deep learning analysis of ECGs. Although some studies have reported that cardiovascular risk [42], susceptibility to certain diseases [43], and personality traits [44] differ according to ABO blood type, the mechanism and causality behind these differences were unclear. The results of our study suggest that the associations between the specific ABO blood type and the CVD risk reported in previous studies are not explainable or contributable by changes in ECGs. This negative result, derived from a large population comprising 124,406 individuals, could provide robust evidence that the two domains do not seem to be directly associated with each other, even though some previous studies have reported their possible association. Finally, there could also be indirect evidence that our models were less likely to be inappropriately overfitted for the dataset.

Recently, wearable devices based on 1-lead or 2-lead ECGs have been increasingly used. Therefore, we attempted to determine the predictive power of the models developed with 1-lead ECGs. Although the data used in the study were simple excerpts of the standard 12-lead supine ECGs, the potential value could be investigated in this manner. Even if the overall performance was lower than when 12-lead ECGs were used, the models with 1-lead ECGs still showed a fair performance in the prediction of age, sex, and BMI.

Several studies have reported ECG analysis models using deep learning [45]. While most studies have focused on arrhythmia, some have proposed models that predict heart failure [46], valvular heart disease [47], and cardiomyopathy [48], which are not possible to diagnose solely based on an ECG. Furthermore, attempts are being made for the extracardiac domain, such as anemia [17] or hyperkalemia [19]. Our model predicted the subjects’ demographic and anthropometric features, providing two potential advantages. First, the age predicted by our model can be used as a functional biomarker that better reflects the overall cardiovascular health [20]. For instance, by comparing the biological age predicted from the ECG and the ‘chronological age’, individuals’ overall health status could be quantified. Second, the model may help in correctly identifying subjects during ECG acquisition. Since subject mismatches can potentially lead to errors in diagnosis or treatment [21,22], our algorithm could provide an opportunity to rectify such errors when the ECG entered deviates considerably from the corresponding subject’s information.

Our study had a few limitations. First, no external validation was performed because we used only a single source of data. Many artificial intelligence models fail to exhibit the same performance on external datasets from different environments. Although ECG data are not likely to carry such a risk as they are gathered using various types of equipment by many operators, further validation using additional datasets is required to generalize the results. Second, all of the data used were acquired in tertiary hospitals. Taking ECG scans in tertiary hospitals may mean that there have been prior indications of particular diseases or conditions, which may result in a deviation of the dataset from the ECG abnormalities and the prevalence rate of accompanying diseases found in the general population. In particular, because BMI was only measured in subjects who underwent cardiac ultrasonography, this problem may be more marked in the BMI prediction model than in the other models. Due to restrictions on sensitive information and data accessibility of the study subjects, we could not confirm the presence or absence of any disease, including heart disease. In future studies, we will include a patient history, including previous heart disease, to enhance the significance of the research. Thirdly, while the ECG is increasingly being utilized outside hospitals through wearable devices with 1-lead ECGs [2,3,4,5], our model was developed using the standard 12-lead supine ECG. It is difficult to apply the algorithm to wearable devices because there are considerable differences in the nature and characteristics of ECGs acquired from wearable devices and conventional monitoring systems [49,50]. Further research on ambulatory ECG monitoring is required to expand our model for general use. Finally, the explainability of the model is limited. The decision making of the model and the reliability and explanation of the rationale for that decision are critical. Therefore, we tried to secure the explainability of the model through a Grad-CAM analysis. As a result, it was found that our model focused on P wave, QRS complex, T wave, and other ECG segments during the prediction process. However, this provided indirect evidence that our model utilized the major components of the ECG. It could not reveal exactly which ECG features the model used for predicting each class. Further studies are required to develop better explainable AI.

## Figures and Tables

**Figure 1 jcm-12-02828-f001:**
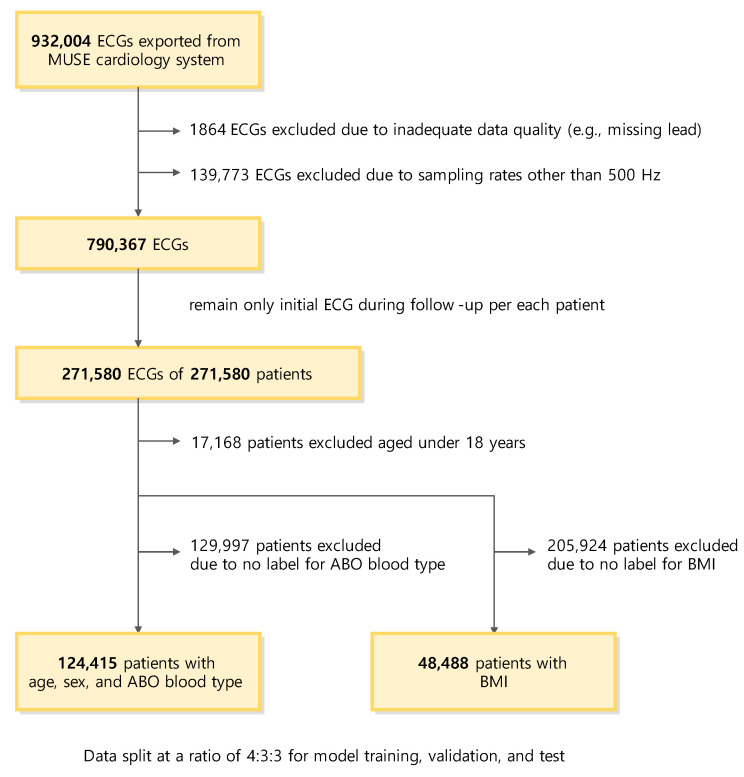
Flow diagram of subject selection and dataset configuration. Abbreviations: ECG, electrocardiography; BMI, body mass index.

**Figure 2 jcm-12-02828-f002:**
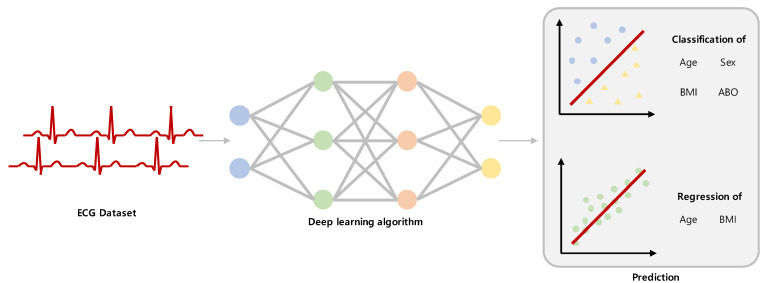
Overview of the study design. A schematic diagram of the research approach. Abbreviations: ECG, electrocardiography.

**Figure 3 jcm-12-02828-f003:**
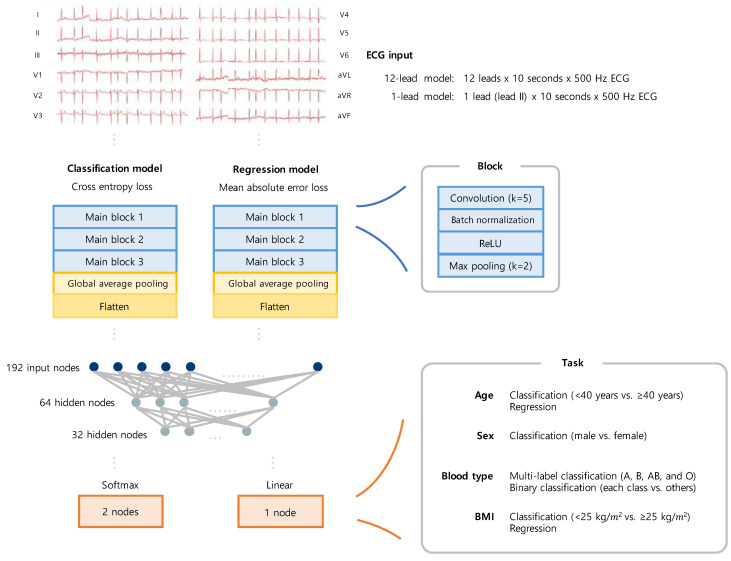
Schematic overview of the models. Structural diagram of the convolution neural network for the tasks. Abbreviations: ECG, electrocardiography; ReLU, rectified learning unit.

**Figure 4 jcm-12-02828-f004:**
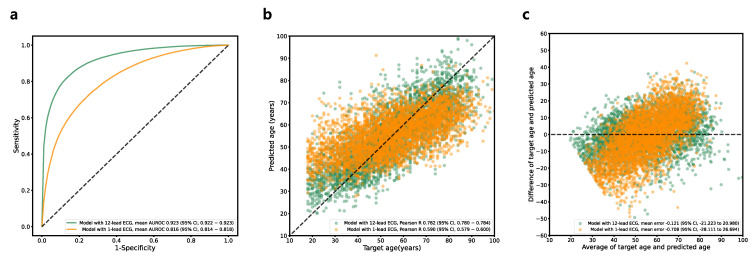
Performance of age prediction model. (**a**) Receiver operating characteristic curve for classification model (<40 years vs. ≥40 years). (**b**) Linear correlation for regression model. (**c**) Bland–Altman plot for regression model. Abbreviations: ECG, electrocardiography; AUROC, area under receiver operating characteristics curve.

**Figure 5 jcm-12-02828-f005:**
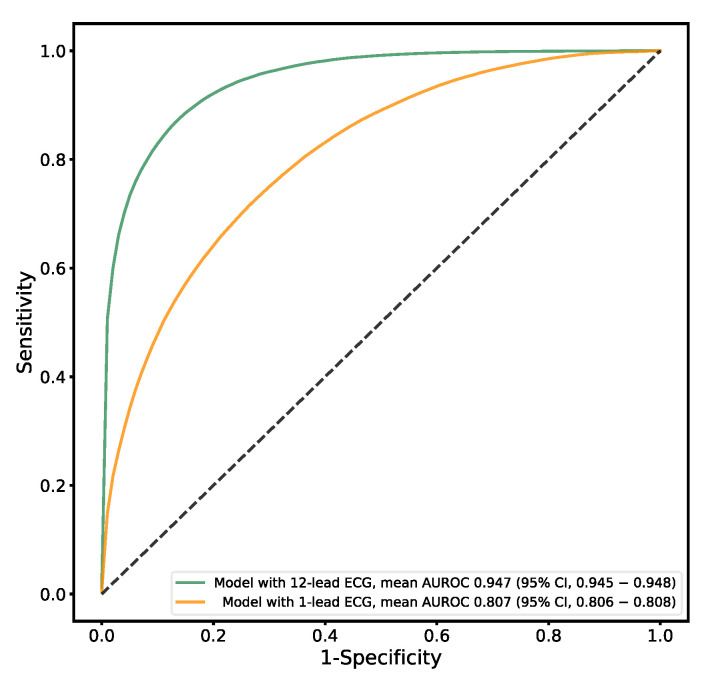
Performance of sex prediction model. Receiver operating characteristic curve (male vs. female sex). Abbreviations: ECG, electrocardiography; AUROC, area under receiver operating characteristics curve.

**Figure 6 jcm-12-02828-f006:**
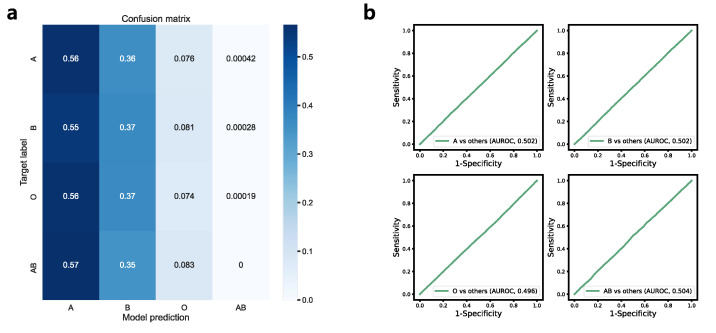
Performance of ABO blood type prediction model. (**a**) Confusion matrix for multilabel classification (A, B, AB, and O) (**b**) Receiver operating characteristic curve for binary classification models. Abbreviations: AUROC, area under receiver operating characteristics curve.

**Figure 7 jcm-12-02828-f007:**
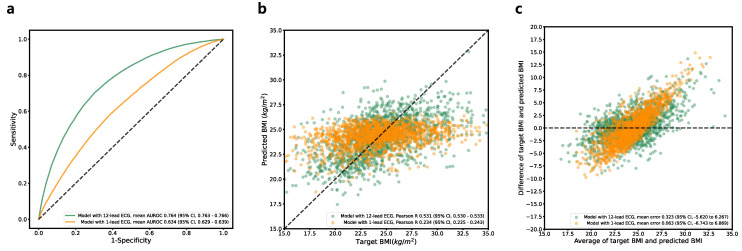
Performance of BMI prediction model. (**a**) Receiver operating characteristic curve for the classification model (BMI of <25 kg/m^2^ vs. ≥25 kg/m^2^). (**b**) Linear correlation for the regression model. (**c**) Bland–Altman plot for the regression model. Abbreviations: BMI, body mass index; ECG, electrocardiography; AUROC, area under receiver operating characteristics curve.

**Figure 8 jcm-12-02828-f008:**
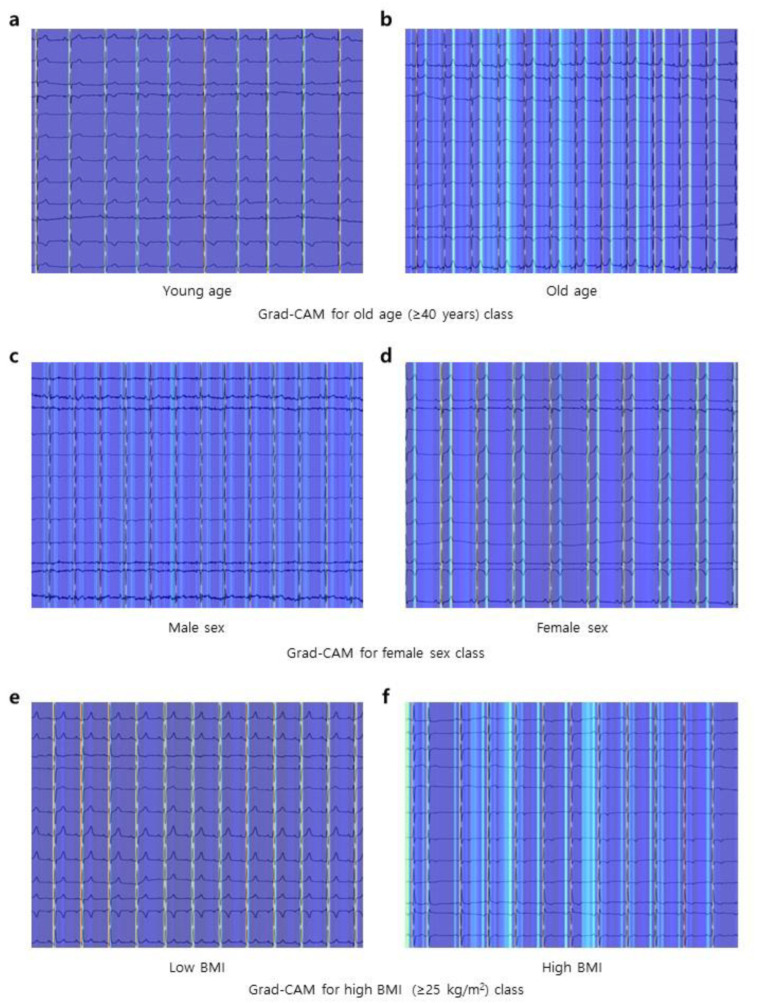
Class activation maps. (**a**,**b**) Grad-CAM for the old age class (≥40 years). The ECG of the young age (<40 years) in panel (**a**) shows minimal attention from the model in predicting target class as old age, whereas the ECG of the old age in panel (**b**) shows strong attention from the model. (**c**,**d**) Grad-CAM for the female sex class. The ECG of the male sex in panel (**c**) shows minimal attention from the model in predicting target class as female sex, whereas the ECG of the female sex in panel (**d**) shows strong attention from the model. (**e**,**f**) Grad-CAM for the high BMI class (≥25 kg/m^2^). The ECG of the low BMI (<25 kg/m^2^) in panel (**e**) shows minimal attention from the model in predicting target class as high BMI, whereas the ECG of the high BMI in panel (**f**) shows strong attention from the model. Abbreviations: Grad-CAM, gradient-weighted class activation map; ECG, electrocardiogram; BMI, body mass index.

**Table 1 jcm-12-02828-t001:** Datasets.

	Training Set	Validation Set	Test Set
**Age (y)**			
No. of unique patients	49,762	37,324	37,329
Mean (SD)	55.25 (17.25)	55.25 (17.24)	55.23 (17.26)
≥40 y, n (%)	39,871 (80.12%)	29,904 (80.12%)	29,908 (80.12%)
<40 y, n (%)	9891 (19.87%)	7420 (19.87%)	7421 (19.87)
**Sex**			
No. of unique patients	49,766	37,324	37,325
Male sex, n (%)	25,432 (51.10%)	19,074 (51.10%)	19,074 (51.10%)
Female sex, n (%)	24,334 (48.89%)	18,250 (48.89%)	18,251 (48.89%)
**ABO blood type**			
No. of unique patients	49,760	37,322	37,324
A, n (%)	15,913 (31.97%)	11,935 (31.97%)	11,935 (31.97%)
B, n (%)	14,362 (28.86%)	10,772 (28.86%)	10,773 (28.86%)
AB, n (%)	14,086 (28.30%)	10,565 (28.30%)	10,566 (28.30%)
O, n (%)	5399 (10.85%)	4050 (10.85%)	4050 (10.85%)
**BMI**			
No. of unique patients	19,393	14,546	14,549
Height (cm), mean (SD)	160.60 (14.94)	160.46 (9.64)	160.62 (20.73)
Weight (kg), mean (SD)	63.08 (13.60)	63.01 (12.14)	62.93 (12.23)
BMI (kg/m^2^), mean (SD)	24.38 (4.36)	24.41 (4.81)	24.75 (33.36)
≥25 kg/m^2^, n (%)	7727 (39.84%)	5812 (39.95%)	5788 (39.78%)
<25 kg/m^2^, n (%)	11,666 (60.15%)	8734 (60.04%)	8761 (60.21%)

Abbreviations: BMI, body mass index.

**Table 2 jcm-12-02828-t002:** Performance metrics for models with standard 12-lead supine electrocardiography.

	Estimation Target			
	Age	Sex	BMI	ABO Blood Type
**Classification model**				
AUROC curve (95% CI)	0.923 (0.922–0.923)	0.947 (0.945–0.948)	0.764 (0.763–0.766)	0.501 (0.496–0.506)
Sensitivity, % (95% CI)	81.96% (81.55–82.37%)	87.42% (85.92–88.91%)	70.00% (67.80–72.20%)	56.12% (4.96–107.27%)
Specificity, % (95% CI)	86.67% (86.27–87.08%)	86.25% (85.07–87.42%)	69.82% (67.30–72.35%)	44.03% (−7.31–95.37%)
PPV, % (95% CI)	95.73% (95.62–95.83%)	85.89% (85.05–86.74%)	60.45% (59.18–61.72%)	25.08% (9.82–40.33%)
NPV, % (95% CI)	56.87 (56.42–57.32%)	87.76% (86.65–88.88%)	77.97% (77.26–78.69%)	74.93% (59.50–90.37%)
**Regression model**				
MAE (95% CI)	8.410 (8.380–8.441)	N/A	2.332 (2.328–2.336)	N/A
Pearson R (95% CI)	0.782 (0.780–0.784)	N/A	0.531 (0.530–0.533)	N/A
R^2^ (95% CI)	0.610 (0.607–0.612)	N/A	0.279 (0.276–0.282)	N/A
ICC (95% CI)	0.636 (0.628–0.645)	N/A	0.474 (0.464–0.484)	N/A

Abbreviations: BMI, body mass index; AUROC, area under the receiver operating characteristic; CI, confidence interval; PPV, positive predictive value; NPV, negative predictive value; MAE, mean absolute error; ICC, intraclass correlation coefficient.

**Table 3 jcm-12-02828-t003:** Performance metrics for models with 1-lead electrocardiography (lead II excerpt).

	Estimation Target		
	Age	Sex	BMI
**Classification model**			
AUROC curve (95% CI)	0.816 (0.814–0.818)	0.807 (0.806–0.808)	0.633 (0.629–0.639)
Sensitivity, % (95% CI)	69.92% (67.41–72.43%)	72.35% (70.41–74.29%)	65.24% (61.76–68.72%)
Specificity, % (95% CI)	77.46% (74.98–79.95%)	72.69% (71.02–74.36%)	53.60% (49.48–57.72%)
PPV, % (95% CI)	91.88% (91.33–92.44%)	71.72% (71.01–72.44%)	48.09% (47.16–49.02%)
NPV, % (95% CI)	41.45% (40.21–42.70%)	73.33% (72.42–74.25%)	70.11% (69.45–70.77%)
**Regression model**			
MAE (95% CI)	11.220 (11.096–11.344)	N/A	2.684 (2.682–2.687)
Pearson R (95% CI)	0.590 (0.579–0.600)	N/A	0.234 (0.225–0.243)
R^2^ (95% CI)	0.345 (0.332–0.358)	N/A	0.048 (0.040–0.056)
ICC (95% CI)	0.428 (0.417–0.440)	N/A	0.107 (0.103–0.111)

Abbreviations: BMI, body mass index; AUROC, area under the receiver operating characteristic; CI, confidence interval; PPV, positive predictive value; NPV, negative predictive value; MAE, mean absolute error; ICC, intraclass correlation coefficient.

**Table 4 jcm-12-02828-t004:** Comparison with previous studies.

	Estimation Target	Method	Training Set	Validation Set	Test Set	AUROC (95% CI)	MAE (95% CI)	R2 (95% CI)
Ours	Age classification	DL	49,762	37,324	37,329	0.923 (0.922–0.923)	-	-
	Age regression	DL	49,762	37,324	37,329	-	8.410 (8.380–8.441)	0.610 (0.607–0.612)
	Sex classification	DL	49,766	37,324	37,325	0.947 (0.945–0.948)	-	-
Z. I. Attia et al. [25]	Age regression	DL	399,750	99,977	275,056	-	6.9 (1.3–15.5)	0.7
	Sex classification	DL	399,750	99,977	275,056	0.968	-	-
E. M. Lima et al. [38]	Age regression	ML	185,444		32,725	-	8.38 (1.38–15.38)	0.71
	Age regression	ML	185,444		14,263	-	8.44 (2.25–14.63)	0.32
	Age regression	ML	185,444		1631	-	10.04 (2.28–17.8)	0.35
H. E. van der Wall et al. [39]	Age regression	ML	6110		118	-	6.9 (1.3–12.5)	0.72 (0.68–0.76)

Abbreviations: DL, deep learning; ML, machine learning; AUROC, area under the receiver operating characteristic; MAE, mean absolute error; CI, confidence interval.

## Data Availability

Not applicable. The data used in this study cannot be disclosed without the permission of the Ethics Committee (irb@yonsei.ac.kr, 82-033-741-1715). The data contain potentially sensitive information such as the subject’s date of birth and gender. Thus, the data are not publicly available because of privacy or ethical restrictions.

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
