# Peer review of "Deep Learning Algorithms for Estimation of Demographic and Anthropometric Features from Electrocardiograms"

_jcm, 2023, doi:10.3390/jcm12082828_

Round 1
Reviewer 1 Report
This study aimed to develop deep learning models using convolutional neural networks (CNNs) to predict demographic and anthropometric factors such as age, sex, ABO blood type, and body mass index (BMI) based on electrocardiogram (ECG) data. The models were developed and validated on a dataset of 124,415 ECGs, achieving high accuracy and AUROC for age and sex estimation, moderate accuracy for BMI estimation, and poor accuracy for ABO blood type estimation. The study suggests that ECGs could be used to estimate individuals' demographic and anthropometric features and develop physiologic biomarkers that better reflect their health status than chronological age.
The manuscript can be improved from the following aspects.
1. Provide more context: While the introduction provides some information about ECG and deep learning, it would be helpful to provide more context about the significance of the problem being addressed. For example, how do accurate predictions of demographic and anthropometric features using ECGs contribute to the field of medicine?
2. Highlight the research gap: The introduction mentions that previous studies have attempted to use deep-learning algorithms to predict diseases using ECGs. However, the research gap that this study is trying to address could be more clearly stated. For example, what is the specific gap in knowledge that this study is addressing?
3. Define key terms: The introduction uses several technical terms, such as deep learning, biosignals, and feature extraction. Defining these terms or providing a brief explanation of what they mean would help readers who are not familiar with these concepts understand the study better.
4. Use more active voice: The introduction uses a lot of passive voice, which can make the text feel less engaging. Using more active voice can help make the text more concise and easier to follow.
5. Clarify the study objectives: While the study aims to develop and validate a deep learning model that predicts demographic and anthropometric features using ECGs, it would be helpful to clarify what the specific objectives of the study are, and how it addresses the research gap mentioned earlier.
6. Provide more details about the study population: The study population should be described in more detail. For instance, the number of patients with cardiac diseases and healthy individuals, as well as the age range of the participants, should be provided.
7. Include a flowchart or diagram to explain the study design: A flowchart or diagram can help readers understand the study design better. This would be particularly helpful for readers who are not familiar with the technical aspects of the study.
8. Describe the hyperparameters used in the deep learning model: The hyperparameters used in the deep learning model, such as the number of layers, batch size, and learning rate, should be described. This will enable readers to replicate the study.
9. Provide more details about the model architecture: The model architecture should be described in more detail. For instance, how were the convolutional layers configured, and what was the purpose of the fully connected layer? This will help readers understand the technical aspects of the model.
10. Provide more details about the data preprocessing: The data preprocessing steps should be described in more detail. For instance, were any filters applied to the ECG signal, and how were the leads combined to generate the 12-lead ECGs?
Author Response
We summarized all the corrections and comments on the feedback and organized them in the response letter. We sincerely thank for leaving such advice and thank you for the opportunity to improve the quality of the manuscript. We hope this revision will supplement our research and make it a more suitable manuscript for journalism.

Reviewer 2 Report
This paper develops a deep learning model based on electrocardiogram (ECG) analysis to predict individuals' age, sex, ABO blood type, and body mass index (BMI) with high accuracy. The research shows that using deep learning models to analyze ECG can accurately predict age and sex. The main clinical value of this study is that using a deep learning model can identify a patient's age and sex during ECG examination. If the model's identification results do not match the data provided by the patient, further verification can be done to reduce diagnostic or treatment errors.
The authors should introduce the pathological relationship between ECG and age, sex, ABO blood type, and BMI in the introduction section and explain relevant research using ECG and machine learning models for estimation.
The authors should provide a table to compare the proposed methods and results with previous studies more clearly.
Author Response

(The authors gave the same response as above.)

Round 2
Reviewer 1 Report
The authors have responded to and addressed all comments received during the review process, ensuring that the manuscript has been thoroughly revised and improved.